Optimizing power allocation for URLLC-D2D in 5G networks with Rician fading channel

Muhammad Owais 1 pingtoowais@gmail.com
Jiang Hong 1 jianghong@swust.edu.cn
Bilal Muhammad 2
http://orcid.org/0000-0002-1449-7340 Muhammad Umer Mushtaq 3
1 School of Information Engineering, Southwest University of Science and Technology , Mianyang, Sichuan , China
2 School of Software Engineering, Northeastern University , Shenyang, Liaoning , China
3 Department of Computer Science, University of Pretoria , Pretoria , South Africa
Alatas Bilal
Electronic publication date: 2025 Feb 18
Publication date: 2025
Volume: 11
Electronic Location ID: e2712
Received 2024 Jul 30; Accepted 2025 Jan 26
Copyright: © 2025 Muhammad et al.
Copyright year: 2025
Copyright holder: Muhammad et al.
License: This is an open access article distributed under the terms of the Creative Commons Attribution License, which permits unrestricted use, distribution, reproduction and adaptation in any medium and for any purpose provided that it is properly attributed. For attribution, the original author(s), title, publication source (PeerJ Computer Science) and either DOI or URL of the article must be cited.
License URL: https://creativecommons.org/licenses/by/4.0/

Keywords: D2D communication, Rician fading, URLLC, Power allocation

Funding: The authors received no funding for this work.

==============================
The rapid evolution of wireless technologies within the 5G network brings significant challenges in managing the increased connectivity and traffic of mobile devices. This enhanced connectivity brings challenges for base stations, which must handle increased traffic and efficiently serve a growing number of mobile devices. One of the key solutions to address these challenges is integrating device-to-device (D2D) communication with ultra-reliable and low-latency communication (URLLC). This study examines the impact of the Rician fading channel on the performance of D2D communication under URLLC. It addresses the critical problem of optimizing power allocation to maximize the minimum data rate in D2D communication. A significant challenge arises due to interference issues, as the problem of maximizing the minimum data rate is non-convex, which leads to high computational complexity. This complexity makes it difficult to derive optimal solutions efficiently. To address this challenge, we introduce an algorithm that is based on derivatives to find the optimal power allocation. Comparisons are made with the branch and bound (B&B) algorithm, heuristic algorithm, and particle swarm optimization (PSO) algorithm. Our proposed algorithm improves power allocation performance and also achieves faster execution with lower computational complexity compared to the B&B, PSO, and heuristic algorithms.

Introduction

The advancements in wireless technology, the number of mobile devices in cellular systems is growing rapidly. As population density in major cities continues to rise, more people are using mobile devices, and the distance between them is shrinking. This has created new possibilities for communication (Banafaa et al., 2023). Technologies such as device-to-device (D2D) communication in 5G have emerged to meet the demand for faster and more efficient information transfer in wireless networks (Papachary, Arya & Dappuri, 2024a). D2D communication eliminates the need for intermediary base stations by enabling direct connection between two devices. It enables devices to create direct connections with one another, facilitating communication without depending on the base station (Borgohain & Choudhury, 2023). D2D communication is an effective solution to address the problem and is being widely embraced in the 5G mobile communications network. Using ultra-reliable and low-latency communication (URLLC) ensures high reliability and low latency for end-to-end communication. Maintaining dependable communication and achieving low latency for optimal control performance is exceedingly challenging. Adopting D2D communication, this approach offers significant advantages, including notable reductions in power consumption, decreased transmission latency, and enhanced overall reliability (Chang et al., 2019; Muhammad et al., 2023). One of the most important components in 5G networks is URLLC and its primary aim is to facilitate the provision of new services and applications that exhibit low latency, availability, and high reliability (Dao et al., 2021). URLLC demands exceptionally stringent criteria, necessitating a 99.999% reliability rate (equivalent to a packet error probability of 10−5) (Zheng, Cheng & Liang, 2023) and 1 ms of end-to-end latency (Li et al., 2023).

Conventional wireless networks have been constructed with a major emphasis on long-packet transmission situations in order to optimize power consumption. However, obtaining high reliability and minimal latency in such scenarios is often complex (Adhikari & Hazra, 2022). This emphasizes the importance of creating unique architectures and transmission strategies to meet the URLLC, which show high reliability as well as low latency. D2D communication under URLLC constraints is highly challenging to achieve stringent reliability and low latency (Palla, Amudala & Budhiraja, 2024). In D2D communications, the short proximity between users allows for power allocation, which is not achievable in traditional cellular communications (Salim, Elsayed & Abdalzaher, 2023). D2D communications are increasingly acknowledged as a promising approach to fulfill the rigorous demands of URLLC (Iqbal et al., 2023; Haque et al., 2023). Allocating resources and optimizing power represent significant challenges in D2D communication within cellular networks. D2D communication improves network capacity, optimizes resource allocation, reduces power consumption, and minimizes latency by leveraging user proximity, a capability not achievable in traditional cellular communications (She & Li, 2023; Alibraheemi et al., 2023). The main objective of this research is to explore the impact of Rician fading on uplink D2D communication, focusing on optimizing power allocation to ensure both reliability and low latency under URLLC constraints.

Contribution

This article focuses on the investigation of Rician fading channels in the propagation of uplink D2D communication in a single-cell cellular environment. System performance in D2D communication can be negatively affected by the influence of a fading environment. Therefore, this study investigates the effect of Rician fading in D2D communication. To address the rigorous QoS demands of D2D communication, the study employs URLLC techniques. The presence of fading in the environment can have an impact on D2D communication, which may lead to a decrease in system performance. It is difficult to solve the minimal rate maximization problem and analysis of the interference. In this study, we find the optimal power allocation for D2D communication by jointly considering reliability and latency constraints in URLLC with low computational complexity. The contributions of this study are outlined as follows: An approach has been developed to optimize power allocation for D2D communication within a cellular network. This formulation addresses the challenges of Rician fading and the stringent requirements of URLLC to enhance overall performance.

The objective is to maximize the minimum rate of D2D users by optimizing power allocation. This problem is non-convex and presents a significant challenge in optimization due to the intricate and non-convex nature of the achievable rate expression.

An iterative algorithm is proposed, based on derivative, to maximize the minimum rate by efficiently finding the optimal power allocation while maintaining a considerably low computational complexity. The performance of the proposed power allocation algorithm is compared with the branch and bound (B&B) algorithm, particle swarm optimization (PSO) algorithm and heuristic algorithm. The computational complexity of the power allocation algorithm is remarkably lower than that of the B&B, PSO, and heuristic algorithms, while it also outperforms these algorithms in meeting the network’s URLLC requirements.

The subsequent sections of this article are structured as follows. The Related Work section presents a overview of prior research conducted by other scholars in the field. The System Model section presents the system model for the proposed approach, while the Optimizing Power Allocation section formulates the power allocation problem. The simulation results are presented in the Simulation Results and Analysis section, and the last section discusses the conclusions and future work.

Related work

Recent studies have concentrated on optimizing resource and power allocation in D2D communication within cellular networks. Huang et al. (2024) propose a value decomposition network for resource allocation in D2D communications underlaying cellular networks. The approach reduces interference and improves spectral efficiency by enabling centralized training and distributed decision-making among D2D users. Similarly, Acharjee, Debnath & Arif (2019) proposed an optimal re-source allocation strategy that prioritizes cellular users while enabling resource sharing with D2D pairs. This approach utilized a PSO algorithm to enhance system throughput and reduce power consumption. However, Hussein, Elsayed & Abd El-kader (2020) introduced an approach that considers the Rician fading and minimizes the overall transmit power of the D2D communication, but they did not provide reliable and low-latency communication. Papachary, Arya & Dappuri (2024b) investigate network slicing in D2D networks to improve Enhanced Mobile Broad-band and URLLC. It optimizes energy efficiency and resource allocation through a Mixed-Integer Nonlinear Program that integrates beamforming and resource sharing selection, using methods like the Dinkelbach approach. Jiang et al. (2024) further explore a full-duplex integrated sensing D2D system within cellular networks. They proposes a joint beamforming and power allocation scheme to enhance the performance D2D networks, formulating a non-convex sum rate maximization problem and introduce a successive convex algorithm introduced to solve this problem.

Some studies have examined the D2D communication in URLLC and attempted to get the high quality-of-service (QoS) requirement in URLLC. Elmadina et al. (2023) proposed an interference avoidance algorithm for power allocation in D2D communication. The proposed algorithm minimize interference between D2D users while optimizing power allocation to enhance communication performance. Similarly, Alruwaili, Kim & Oluoch (2024) introduced a method for optimizing power allocation and throughput in 5G cellular systems using D2D communication and a modified Gale-Shapley algorithm. Veedu & Manjappa (2024) proposed frameworks for many-to-many resource allocation and optimal power control in underlay cellular D2D communication to enhance public safety applications. These proposed methods improve the system sum rate and power control. Sanusi & Nasr (2024) introduced a matching theory approach for resource allocation in D2D communication to meet URLLC requirements. Similarly, Kai et al. (2023) introduced a reinforcement learning approach for power allocation in D2D networks. A mode-switching scheme was introduced to optimize power allocation in D2D communications within 5G networks, aiming to maximize joint channel capacity (Gao et al., 2023). Chang et al. (2021) addressed the optimization problem of transmit power in D2D communication for real-time wireless control systems with URLLC and control requirements, a probability-based D2D technique that uses Rayleigh fading, which is less efficient than Rician fading in reducing power consumption.

System model

We investigate uplink D2D communication in a single-cell network of a cellular system, as illustrated in Fig. 1. The cellular base station (BS) is located in the center of the cell, and the cellular users (CUs) are uniformly distributed across the entire cell area. We consider that there are M D2D pairs and N cellular users.

Figure 1 System model.

The sets of cellular users and D2D pairs are denoted by L={1,2,…,N} and D={1,2,…,M} where M≤N. The resource reuse indicator, denoted as pn,m∈{0,1}. Specifically, pn,m=1 if D2D pair m reuses the channel of cellular user n, otherwise pn,m=0 (Xu et al., 2021). The known constants, assuming slow fading components, while the fast-fading components are represented by random variables. Gmd and Gn,mcd are fast-fading coefficients. The interference caused by other D2D connections is denoted as Gn,mcd and follows an exponential distribution with a mean of zero and a unit variance. The D2D transmitter (DT) m toward the receiver is expressed as Smd is slow fading coefficient, and the power gain of the interference channel between the cellular user n and the D2D receiver (DR) m is denoted as Sn,mcd. Due to strong line-of-sight (LoS) components, we model the communication channel using a Rician fading model, which is suitable for environments where a dominant direct LoS signal exists along with scattered multipath components. The probability density function (PDF) of Gmd is show in Eq. (1), The Rician K-factor quantifies the ratio of the power in the LoS component to the scattered components, with higher values indicating a stronger LoS path and I0 is the zeroth-order Bessel function (Peng et al., 2014; Zhang et al., 2024).

(1) fGmd(z)=(1+K)e−Kz¯exp(−(1+K)zz¯)I0(2K(1+K)zz¯).

Due to their shared spectrum of resources, D2D users encounter interference from cellular users as well as other D2D pairings. To assess the signal quality, we calculate the SINR, which measures the power of the desired signal relative to interference and noise. A higher SINR value indicates better signal reception (Pan & Zheng, 2023). The Eq. (2) expresses the SINR for the mth D2D link. Pnc denotes the transmit power of cellular connection n, and the transmit power of D2D link m is represented as Pmc. The additive noise power is denoted as σ2.

(2) γmd=PmdSmdGmd∑n∈L⁡pn,mPncSn,mcdGn,mcd+σ2.

In scenarios involving URLLC, achieving low latency requires users to transmit short packets. The D2D communication achievable rate is given as:

(3) Rm=BmIn2[Cm−VmTmBmfQ−1(ϵm)]

where Bm represents the bandwidth and Vm corresponds to the channel dispersion. Tm refers to the transmission time delay, and Cm represents the Shannon capacity. fQ−1(⋅) is the inverse of the Q function and ϵm is the transmission error probability. The Shannon capacity is formulated as follows, relying on the received SINR.

(4) Cm=log(1+γmd)

where Vm is the channel dispersion (Chang et al., 2019), it can be expressed as

(5) Vm=(1−1(1+γmd)2)≈1

When SINR exceeds 5dB then Vm≈1 (Sun, She & Yang, 2023). The analytical expression for the outage probability of D2D communication is given as follows.

(6) PmOP=Pr[γmd≤γ0=γdγI≤γ0orγd≤γth].

The minimal instantaneous signal power threshold is denoted as γth, whereas the minimal SINR threshold is represented as γ0. The signal power, represented as γd, where γd=PmdSmdGmd and the total interference power, denoted as γI, where γI=∑n∈Lpn,mPncSn,mcdGn,mcd. The symbol γI encompasses both co-tier and cross-tier interference arising from interference among interference from DTs directed toward other DRs and cellular users. LI is the sum of the instantaneous powers, equal to the total instantaneous interference power, which is represented as γI, where γI=∑i=1LI⁡γi (Huq, Mumtaz & Rodriguez, 2016). The D2D outage probability can be formulated according to Eq. (6) as follows:

(7) PmOP={γmd≤γ0}=Pr(PmdSmdGmd∑n∈L⁡pn,mPncSn,mcdGn,mcd≤γ0).

The power of interference from the cellular user to the D2D receiver is generally much greater than the power of noise (Peng et al., 2014). We assume that D2D links are interference-limited, and we can ignore the influence of noise power on the outage probability (Yin et al., 2015). Equation (7) is represented in the form of a PDF as follows:

(8) PmOP=1−∫γth∞(∫0γdγ0⁡fγI(γI)dγI)fγd(γd)dγd.

The PDF of the instantaneous signal power is represented as fγd(γd), and the total interference power PDF is represented as fγI(γI). The received signal from the intended user conforms to a Rician distribution, and there is Rayleigh interference in the system that is LI i.i.d. The total instantaneous interference power γI PDF can be denoted as

(9) fγI(γI)=γILI−1γ¯ILI(LI−1)!exp(−γIγ¯I).

The statistical average of γI is denoted as γ¯I, and the instantaneous signal power γd PDF denoted as

(10) fγd(γd)=(K+1)γ¯de[−K−(K+1)γdγ¯d]I0(2K(K+1)γdγ¯d).

By applying Eqs. (9) and (10), the outage probability of D2D can be reformulated within Eq. (8)

(11) PmOP=1−∫γth∞(∫0γdγ0⁡γILI−1γ¯ILI(LI−1)!exp(−γIγ¯I)dγI)((K+1)γ¯de[−K−(K+1)γdγ¯d]I0(2K(K+1)γdγ¯d)).

We obtain the outage probability, by solving the inner integral of Eq. (11). This indicates that the outage probability is influenced by the Rician K-factor, the signal power, and the total interference power γI. Increasing the strength of the LoS component (higher K-factor) or reducing the total interference can lower the outage probability, thereby enhancing communication reliability.

(12) PmOP=exp[−K+K(1+γ¯d(K+1)γ¯d)](1+γ¯d(K+1)γ0γ¯I).

Proof: See Appendix A.

The ergodic capacity of Cm, derived from the overall performance across all channel fading states, can be expressed as follows

(13) Cm=E[log(1+γmd)]

(14) E[log(1+γmd)]=∫0∞⁡log(1+x)fγmd(x)dx.

Over the fast-fading distribution, the expectation E[⋅] is taken. By applying integration-by-parts

(15) ∫x=0∞⁡∫y=0x⁡11+yfγmd(x)dydx

(16) ∫y=0∞⁡11+ydy∫x=y∞⁡fγmd(x)dx.

The ergodic capacity is calculated by the following expression:

(17) Cm∗=∫0∞⁡1−Fγmd(x)1+xdx.

where Fγmd(x)=Pr(γmd≤γ0) is provided in Eq. (12), then Eq. (3) can be reformulated as:

(18) Rm∗=Bmln2[Cm∗−VmTmBmfQ−1(ϵm)].

Equation (19) provides an expression for the probability of a packet error, hm=Rm∗Tm represents the amount of bits that must be transferred during each transmission.

(19) ϵm∗=fQ{TmBmVm[Cm∗−hmln2TmBm]}.

The following constraints must be satisfied in order to satisfy URLLC reliability requirements.

(20) ϵm∗≤ϵmax.

The maximum packet error probability denoted as ϵmax, is restricted by the Quality of Service requirements for URLLC and additionally imposes a constraint on the time delay for communication such as

(21) Tm≤Tmax.

The successful transmission probability can be formulated by considering the reliability constraint in URLLC such as

(22) 1−ϵm∗≥1−ϵmax.

The transmit power that meets the constraints can be derived from Eq. (3) as follows

(23) Rm=Bmln2[Cm−1TmBmfQ−1(ϵm∗)]

(24) RmTmln2TmBm=Cm−VmTmBmfQ−1(ϵm∗)

(25) Cm=hmln2TmBm+VmTmBmfQ−1(ϵm∗)

where Cm=log(1+γmd)

(26) log⁡(1+PmdSmdGmd∑n∈L⁡pn,mPncSn,mcdGn,mcd+σ2)=hmln2TmBm+1TmBmfQ−1(ϵm∗)

(27) 1+PmdSmdGmd∑n∈L⁡pn,mPncSn,mcdGn,mcd+σ2=exp[hmln2TmBm+1TmBmfQ−1(ϵm∗)]

(28) Pmd=∑n∈L⁡pn,mPncSn,mcdGn,mcd+σ2SmdGmd{exp[hmln2TmBm+1TmBmfQ−1(ϵm∗)]−1}

where ∑n∈L⁡pn,mPncSn,mcdGn,mcd+σ2SmdGmd solved by outage probability Eq. (7)

(29) Pmd=PmOP{exp[hmln2TmBm+1TmBmfQ−1(ϵm∗)]−1}.

Equation (29) represents the transmit power Pmd required to meet certain constraints while considering the outage probability PmOP in D2D communication. It shows the relationship between transmit power, the required signal quality, and the interference experienced in the system.

Optimizing power allocation

In this section, we present the optimization problem with the objective of maximizing the minimum achievable rate for D2D users, with a specific emphasis on optimizing power allocation. The formulation is presented as follows:

P1:maxminPmdm{Rm∗}

(30a) s.tϵm∗≤ϵmax,∀m∈D

(30b) Tm≤Tmax,∀m∈D

(30c) Pr{γmd≤γ0}≤δ0,∀m∈D

(30d) ∑m∈DPmd≤Pmaxd.

This optimization problem is to find the total optimal transmit power ∑m∈DPmd, in order to maximize the minimum achievable rate. Equation (30a) is utilized to ensure the D2D users’ reliability. The utilization of the constraint expressed in Eq. (30b) aims to enforce a limitation on the transmission time delay, ensuring that it does not exceed the maximum transmission time delay Tmax. In Eq. (30c), δ0 represents the maximum allowable outage probability constraint and, Eq. (30d) defines the transmit power constraint, which imposes limitations on the total transmit power. Pmaxd denotes the maximum transmit power.

We minimize the total power consumption ∑m∈DPmd. The following problem can be solved to obtain the minimum total transmit power.

P2:maxminPmdm{Rm∗}

(31a) s.tϵm∗≤ϵmax,∀m∈D

(31b) Tm≤Tmax,∀m∈D

(31c) Pmd≥0,∀m∈D

(31d) Pr{γmd≤γ0}≤δ0,∀m∈D

(31e) ∑m∈DPmd≤Pmaxd.

Equation (31c) guarantees that each user is allocated a non-negative power. Due to interference, the achievable rate is non-convex, as shown in Eqs. (2) and (18), which makes the power allocation difficult to solve. We use an algorithm based on derivative to address the non-convex function in order to overcome this challenge. The best possible solution can be reached through the resolution of the provided set of equations.

(32) Rm≜fm(Pmd∗)=Bmln2[Cm−1TmBmfQ−1(ϵm∗)]

(33) dfm(Pmd∗)dPmd∗=Bmln2(PmOP∗1+PmOP)

Equation (33) represents the derivative of the function fm(Pmd∗) with respect to the transmit power Pmd∗, is provided in Appendix B.

Proposed power allocation optimization algorithm

The proposed power allocation algorithm, based on derivative is used to identify the optimal solution for the power allocation problem in D2D communication while maintaining the QoS requirements of URLLC. The algorithm is designed to ensure low computational complexity. The flowchart in Fig. 2 presents an iterative algorithm for optimizing power allocation. The algorithm begins by calculating the initial power allocation for each D2D device using Eq. (29) and sets this as a starting point. The algorithm then iteratively adjusts the power allocation based on the difference between total allocated power and the maximum limit, using a step size α controls the adjustment to the power values in each iteration and derivative βm to guide the adjustments. The loop continues until the total power is within a tolerance Δ, ensuring an optimal solution with low computational complexity. The following key steps are used to describe the power allocation algorithm:

Figure 2 Flowchart of the proposed power allocation optimization algorithm.

In steps 1–3, Calculate the Pmd by using Eq. (29) and set to Pmd∗. Subtract the maximum power, Pmaxd from the sum of the calculated power, ∑m∈DPmd and assign it to l as follows l=|Pmaxd−∑m∈DPmd|.

In step 4, α=lY, where α is the adjustment step of Pmd∗ and Y is the controlling factor for α and also controls the performance and computational complexity. l represents the difference between the maximum allowable power Pmaxd and the sum of the current power values. Allowing us to assign an appropriate value based on the specific practical requirements and the lower bound feasible region is set to ωlb=0.

In step 5, we initialize a variable βm, to store the derivative of |fm′(Pmd∗)|. This variable is utilized in step 8 to select the suitable Pmd∗. It may also undergo updates in step 11 or step 13 during each iteration of the algorithm.

In steps 6, to decide when to end the loop, we set a tolerance threshold Δ rather than use a counter, which determines when to stop the algorithm. The loop will continue adjusting the power allocation until the difference between Pmaxd and the total allocated power is less than Δ.

In steps 7–8, if during the current iteration, the selected Pid∗ exceeds the feasible region even after adjustment, then it needs to select another Pmd∗ for adjustment after this iteration. The loop continues as long as the total power allocation difference Pmaxd−∑m∈DPmd∗ is greater than Δ.

In steps 9–11, provided the feasible region to optimize the Pid∗ and instead of updating every βm for m∈D and m≠i, we update only βi.

In step 13, if the value of Pid∗ exceeds the feasible region after adjustment, we assign an infinite to βi to ensure that i will not be selected again in step 8, this can prevent the possibility of an infinite loop. The power allocation algorithm solves the problem P2, as shown in the power allocation algorithm.

Algorithm 1 Power allocation optimization algorithm.

Input: Δ,Pmaxd,ωlb	
Output: Pmd∗	
 1: Calculate the Pmd by using Eq. (29);	
 2:  Pmd∗=Pmd;	
 3:  l=|Pmaxd−∑m∈DPmd|;	
 4:  α=lY;	
 5:  βm=|fm′(Pmd∗)|;	
 6: Initialize the tolerance threshold Δ to control the loop;	
 7: while |Pmaxd−∑m∈DPmd∗|≥Δ do	
 8:     i=arg⁡max{βm};	
 9:    if (Pid∗−α<PmaxdandPid∗−α>0)orPid∗−α>ωlb then	
10:        Pid∗=Pid∗−α;	
11:        βi=|fi′(Pid∗)|;	
12:    else	
13:        βi=−∞;	
14:    end if	
15: end while	

Complexity analysis

In the proposed power allocation algorithm, computational complexity is determined by the adjustments made in step 9 and step 13 from the power allocation algorithm, if in the current iteration, the selected Pid∗ exceeds the feasible region after adjustment, βi will be assigned to an infinite. This ensures that the same Pid∗ will not be selected again, as we always choose the i value with the maximum βi in step 8. Consequently, for a specific i, the occurrence of the currently chosen Pid∗ being unable to be adjusted due to the constraints of the feasible region will occur at most, only once. Therefore, in the ideal scenario where all the chosen Pid∗ can be adjusted during the iterations, the constraint in P2 will be fulfilled after adjusting Pid∗ forY times. Therefore, the power allocation algorithm computational complexity is O(Y).

Simulation results and analysis

In this section, we examine and analyze the performance of the proposed power allocation algorithm, conducting a comparative evaluation against the B&B, PSO and heuristic algorithms. The objective of the optimization problem is to maximize the minimum rate through efficient power allocation while considering Rician fading conditions. A key parameter in our analysis is the URLLC QoS, which ensures that stringent requirements for reliability and latency are met, along with the K-factor, representing the ratio of direct LoS signal power to multipath components. Higher K-factors, with stronger LoS, improve reliability and data rates, while lower K-factors increase fading and reduce rates. The power allocation algorithm focuses on minimizing power consumption while meeting the requirements of URLLC. By selecting a range of K-factors, from high to low, we evaluate the power allocation algorithm effectiveness in optimizing power allocation in both ideal and challenging channel conditions. The simulation parameters we used are summarized in Table 1.

Table 1 Simulation parameters.

Parameters	Values	
K	1,2,4,6,8,10	
Cell radius	500m	
D2D distance (min, max)	[15m,50m]	
LN	10	
DM	8	
Noise spectral density	−174dBm/Hz	
Pmaxd	10dBm	
Bmax	20MHz	
Shadowing standard deviation	10dB	
Tmax	0.1ms	
Δ	1×10−3	
Cellular D2D links path loss model	148+40log(d)	
ϵmax	1×10−5	
δ0	1×10−2	

Figure 3 shows that the achievable rate for D2D communication improves as the Rician K-factor increase. This improvement is due to the robust LoS connection and reduced multipath propagation loss. When the K-factor is high, D2D users achieve higher data rates as interference is minimized.

Figure 3 D2D communication achievable rate with various K factors (higher K-factors providing better performance as the number of users increases).

Figure 4 compares the proposed power allocation algorithm transmit power performance with the B&B, PSO and heuristic algorithms. The evaluation was conducted using various Rician K-factors. The results show that the power allocation algorithm delivers the best solution while consuming the least amount of power in D2D communication. Simultaneously, a notable performance gap is observed between the proposed power allocation algorithm, the PSO, the heuristic algorithm, and the B&B algorithm. The optimization of transmit power takes place as the Rician K-factor increases. This is due to a higher K-factor indicates a stronger LoS component and reduced signal propagation loss resulting from multiple signal paths.

Figure 4 Optimal D2D transmission power under URLLC with various Rician K-factors, comparing PSO, B&B, heuristic and the proposed power allocation algorithm.

The average computation time (ACT) and average transmit power (ATP) for the proposed power allocation algorithm, the B&B, PSO algorithm and heuristic algorithm with various Rician K-factors are shown in Table 2. The power allocation algorithm attains optimal outcomes while maintaining lower computational complexity. The comparative analysis indicates that the proposed power allocation algorithm outperforms the B&B algorithm, PSO algorithm, and heuristic algorithm in terms of speed while also achieving optimal power allocation. Moreover, the heuristic and PSO algorithms, despite their low computational complexity, exhibits faster performance compared to the B&B algorithm.

Table 2 ATP (w) and ACT (s) for the proposed power allocation algorithm, B&B algorithm, PSO algorithm and heuristic algorithm.

	Proposed algorithm	B&B algorithm	PSO algorithm	Heuristic algorithm	
K	ATP	ACT	ATP	ACT	ATP	ACT	ATP	ACT	
1	3.14×10−4	5.73	3.31×10−4	11.31	3.41×10−4	9.15	3.61×10−4	7.38	
2	2.91×10−4	5.63	3.05×10−4	11.19	3.13×10−4	9.37	3.23×10−4	7.33	
4	2.66×10−4	5.71	2.75×10−4	11.39	3.83×10−4	9.12	2.88×10−4	7.35	
6	2.32×10−4	5.66	2.45×10−4	11.64	2.53×10−4	9.13	2.61×10−4	7.32	
8	1.98×10−4	5.65	2.11×10−4	11.58	2.24×10−4	9.29	3.30×10−4	7.28	
10	1.59×10−4	5.62	1.72×10−4	11.13	1.84×10−4	9.17	1.92×10−4	7.17	

Figure 5 illustrates the convergence behavior of the proposed power allocation algorithm, the PSO algorithm, the heuristic algorithm, and the B&B algorithm. The minimal transmit power for D2D communication achieved by the power allocation algorithm exhibits a rapid decrease in the number of iterations, in comparison to the B&B algorithm and other algorithms. The gap of the power allocation algorithm decreases much more rapidly compared to the other algorithm. Figure 6 compares the transmit power performance of Rayleigh fading with the proposed power allocation algorithm, B&B, PSO, and heuristic algorithms. The proposed algorithm achieves the best performance, consuming the least power in D2D communication due to the advantages of Rician fading. Rayleigh fading performs worse due to the non-line-of-sight.

Figure 5 Comparison of transmit power convergence among the B&B algorithm, the PSO algorithm, the heuristic algorithm, and the proposed power allocation algorithm.

Figure 6 Transmit power comparison for D2D users using Rayleigh, heuristic, PSO, B&B, and proposed power allocation algorithms.

Figure 7 illustrates the comparison of average transmit power between the proposed power allocation algorithm, the B&B algorithm, the PSO algorithm, and the heuristic algorithm as a function of the Rician K-factor. The Rician K factor characterizes the ratio between the LoS signal and the scattered signals in Rician fading environments. As the K factor increases, indicating stronger LoS conditions the average transmit power decreases. However, the power allocation algorithm consistently outperforms the B&B, PSO and heuristic algorithms, requiring less transmit power across all K values. This shows the efficiency of the power allocation algorithm, particularly at higher K factors, where the performance gap between the proposed power allocation algorithm and the B&B, heuristic and PSO algorithms indicates a more power-efficient solution in environments with strong LoS components.

Figure 7 Comparison of average transmit power as a function of the Rician K-factor for the proposed power allocation algorithm, B&B algorithm, PSO algorithm, and heuristic algorithm.

Figure 8 compares the average computation time for different power allocation algorithms as a function of parameter K. The proposed algorithm achieves the lowest average computation time across all values of K, highlighting its computational efficiency compared to other methods. B&B exhibits the highest computation time, followed by PSO and heuristic algorithms, respectively, indicating their relative computational inefficiencies.

Figure 8 Comparison of average computation time as a function of the Rician K-factor for the proposed power allocation algorithm, B&B algorithm, PSO algorithm and heuristic algorithm.

The Fig. 9, demonstrates that as the Rician K-factor increases, the outage probability of D2D communication improves. Because of the robust LoS communication and reduced multi-path propagation loss. We see that the quality of communication greatly increases due to the LoS between D2D communications, resulting in increased data rates for users.

Figure 9 Outage probability for D2D communication (lower outage probability with higher K-factors as the number of users increases).

Figure 10 compares the average transmit power performance of the proposed power allocation algorithm, B&B, PSO, and heuristic algorithms across different values of K under URLLC requirements. The power allocation algorithm consistently demonstrates lower transmit power consumption compared to the other algorithms, highlighting its efficiency in optimizing power allocation in D2D communication scenarios with URLLC constraints. The power allocation algorithm performs better than the B&B, PSO and heuristic algorithms in terms of overall performance. Our algorithm minimizes power consumption and maintains low computational complexity for D2D communication in a URLLC scenario under the Rician fading, demonstrating the effectiveness of our approach.

Figure 10 Optimal average D2D transmit power under URLLC with various Rician K-factors.

Conclusion

In this paper, we have proposed an algorithm based on derivative to optimize the power allocation in D2D communication. This research focused on analyzing the uplink D2D communication in a single-cell environment within the cellular system, with a particular emphasis on investigating the characteristics of the Rician fading. We defined an optimization problem related to power allocation, with the objective of maximizing the minimal achievable rate in D2D communication. To solve the non-convex optimization problem, we introduced an algorithm based on a derivative that iteratively converges toward the optimal solution. Our proposed power allocation algorithm aims to optimize power allocation in D2D communication while considering the URLLC with Rician fading. The power allocation algorithm iteratively reaches an optimal solution and is then subjected to a comparison with the B&B, heuristic, and PSO algorithms. The power allocation algorithm exhibits notably lower computational complexity compared to the B&B algorithm, heuristic algorithm, and PSO algorithm. Future research can explore improving power allocation using advanced machine learning techniques, such as reinforcement learning, for more adaptive solutions. This study focuses on a single-cell environment, and future research will examine the algorithm in multi-cell environments with various fading models. Extending it to 6G and IoT systems for resource block allocation, and interference management, would address the growing demands for low latency and high reliability in future wireless networks.

Appendix A

From Eq. (7)

(34) PmOP={γmd≤γ0}=Pr(PmdSmdGmd∑n∈L⁡pn,mPncSn,mcdGn,mcd≤γ0)

(35) PmOP=1−∫γth∞(∫0γdγ0⁡fγI(γI)dγI)fγD(γd)dγd

Eqs. (9) and (10) in Eq. (35) can be used to rewrite the D2D outage probability

(36) PmOP=1−∫γth∞(∫0γdγ0⁡γILI−1γ¯ILI(LI−1)!exp(−γIγ¯I)dγI)((K+1)γ¯de[−K−(K+1)γdγ¯d]I0(2K(K+1)γdγ¯d))

Solving Eq. (36) inner integral and obtaining the outage probability as

(37) PmOP=1−Q1(2K,2(K+1)γthγ¯d)+K+1γ¯d∑j=0LI−11j!∫γth∞(γdγ0γ¯I)jexp[−K−(1γ0γ¯I+K+1γ¯d)γd]I0(2K(K+1)γdγ¯d)dγd

Q1(⋅,⋅) denotes the first-order Marcum Q-function. The outage probability can be expressed by solving the integral and using the Q-function to simplify the Eq. (37) (Nuttall, 1975).

(38) PmOP=1−Q1(2K,2(K+1)γthγ¯d)+a22K∑j=0LI−1ejj!Q2j+1,0(a,q)

where ej=exp[−K+K(1+γ¯d(K+1)γ0γ¯I)](2+2(K+1)γ0γ¯Iγ¯d)j, a=2K(1+γ¯d(K+1)γ0γ¯I) and q=2(1+K+γ¯dγ0γ¯I)γ0γ¯d. Where LI=1 (Yang & Alouini, 2002; Yao & Sheikh, 1990), the Eq. (38) can be expressed as

(39) PmOP=1−Q1(2K,2(K+1)γthγ¯d)+a22Kexp[−K+a22]Q1(a,q)

The interference-limited γth=0, the Eq. (39) can be

(40) PmOP=exp[−K+K(1+γ¯d(K+1)γ¯d)](1+γ¯d(K+1)γ0γ¯I)

Appendix B

The derivation of Pmd∗ is as follows

(41) Rm=Bmln2[Cm−1TmBmfQ−1(ϵm∗)]

(42) Rm≜fm(Pmd∗)=Bmln2[Cm−1TmBmfQ−1(ϵm∗)]

(43) dfm(Pmd∗)dPmd∗=ddPmd∗(Bmln2[Cm−1TmBmfQ−1(ϵm∗)])

where Cm=log(1+γmd)

(44) dfm(Pmd∗)dPmd∗=ddPmd∗(Bmln2[log(1+Pmd∗SmdGmd∑n∈L⁡pn,mPncSn,mcdGn,mcd+σ2)−1TmBmfQ−1(ϵm∗)])

(45) dfm(Pmd∗)dPmd∗=Bmln2(ddPmd∗[log⁡(1+Pmd∗SmdGmd∑n∈L⁡pn,mPncSn,mcdGn,mcd+σ2)]+ddPmd∗[−1TmBmfQ−1(ϵm∗)])

(46) dfm(Pmd∗)dPmd∗=Bmln2([11+Pmd∗SmdGmd∑n∈L⁡pn,mPncSn,mcdGn,mcd+σ2]ddPmd∗[1+Pmd∗SmdGmd∑n∈L⁡pn,mPncSn,mcdGn,mcd+σ2]+0)

(47) dfm(Pmd∗)dPmd∗=Bmln2(SmdGmd∑n∈L⁡pn,mPncSn,mcdGn,mcd+σ21+Pmd∗SmdGmd∑n∈L⁡pn,mPncSn,mcdGn,mcd+σ2)

where PmOP∗={wmd≤w0}=wdwI≤w0=Pr(SmdGmd∑n∈L⁡pn,mPncSn,mcdGn,mcd+σ2≤w0), Where wd=SmdGmd and wI=∑n∈L⁡pn,mPncSn,mcdGn,mcd. Where PmOP∗ and Pmd∗SmdGmd∑n∈L⁡pn,mPncSn,mcdGn,mcd+σ2 solve by outage probability Eq. (7).

(48) dfm(Pmd∗)dPmd∗=Bmln2(PmOP∗1+PmOP).

Supplemental Information

Supplemental Information 1 D2D Matlab Code.

Additional Information and Declarations

Competing Interests

The authors declare that they have no competing interests.

Author Contributions

Owais Muhammad conceived and designed the experiments, performed the experiments, analyzed the data, performed the computation work, prepared figures and/or tables, authored or reviewed drafts of the article, and approved the final draft.

Hong Jiang conceived and designed the experiments, authored or reviewed drafts of the article, and approved the final draft.

Muhammad Bilal conceived and designed the experiments, authored or reviewed drafts of the article, and approved the final draft.

Mushtaq Muhammad Umer conceived and designed the experiments, authored or reviewed drafts of the article, and approved the final draft.

Data Availability

The following information was supplied regarding data availability:

The code is available in the Supplemental File.

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
