# Peer review of "Optimizing power allocation for URLLC-D2D in 5G networks with Rician fading channel"

_PeerJ Computer Science, doi:10.7717/peerj-cs.2712_

## Round 0.1 · original submission · Major Revisions

Based on my evaluation upon receiving reviewers' comments, it is apparent that there are critiques to be addressed by the authors. Thereby, I provide an opportunity to carefully address the comments and submit a revision.

·

Basic reporting

Although the overall presentation of the work is good, the literature overview of the work is weak. Also the ratio of recently published (2022-) work must be more than 25% with at least a few citations from 2024.
General spelling errors will be useful in maintaining a grammar check of the article.

Experimental design

no comment

Validity of the findings

Authors must add a brief future work section or at least a paragraph at the end of the section conclusion, what possible works can be continued from the results of this paper and what authors will research after this paper.

Additional comments

The authors select an interesting topic. The proposed work addresses an algorithm based on derivatives to find the optimal power allocation. Although the presentation is generally good, some of them need to be disrupted.

Although the overall presentation of the work is good, the literature overview of the work is weak. Also the ratio of recently published (2022-) work must be more than 25% with at least a few citations from 2024.
General spelling errors will be useful in maintaining a grammar check of the article.
Authors must add a brief future work section or at least a paragraph at the end of the section conclusion, what possible works can be continued from the results of this paper and what authors will research after this paper.

Reviewer 2 ·

Basic reporting

In general, the journal is meet the requirement as it consists of novelty

Experimental design

The experimental design and methodology are acceptable

Validity of the findings

The comparison has been implemented

Additional comments

In general, the journal is meet the requirement as it achieved the objectives

Reviewer 3 ·

Basic reporting

The simulation contains only three simulation results which is not convincing at all.
The abstract is vague. What is the problem addressed in the proposed article.
the contributions of the article is not mentioned.
How the parameters are selected is not explained.
What is the current status of the problem.
limitations and the future work are not properly explained.

Experimental design

The simulation contains only three simulation results which is not convincing at all.

How the parameters are selected is not explained.

Validity of the findings

I am not sure

Additional comments

The paper cannot be accepted in the current form,

·

Basic reporting

1 Improve the clarity of certain sections by revising awkward sentences and ensuring that professional, unambiguous English is used consistently throughout.
2 Ensure that all references are up to date. Incorporate recent literature (2022-2023) to provide better context and background.
3 The structure is mostly well-organized. However, some sections, such as the algorithm explanation, need more detailed descriptions to improve readability.
4 The definitions of technical terms (e.g., SINR, Rician fading) and key equations (e.g., Equation 12) need more detailed explanations for clarity.
5 While the figures are adequate in quality, enhance the captions to make them fully self-contained with clear descriptions of axes and parameters.

Experimental design

1 Clarify the research question and emphasize how it fills the identified knowledge gap. While this is somewhat clear, it can be stated more explicitly in the introduction.
2 Expand on the description of how the chosen parameters (e.g., K-factors) impact the results and why these values were selected.
3 Ensure that the methods, particularly the algorithm, are described in sufficient detail so that they can be replicated by readers. Add explanations for the key parameters (e.g., step size, tolerance) to support replication.

Validity of the findings

1 The data and simulations provided are sound but would benefit from more explanation regarding the choice of simulation parameters and their impact on the results.
2 Ensure that all underlying data for the experiments are fully described and justified to demonstrate robustness and statistical soundness.
3 The conclusions are well-linked to the research question but expand the discussion on broader implications and possible limitations of the algorithm. Also, suggest potential future research directions.

Additional comments

none

---

## Round 0.2 · Major Revisions

I have received a set of review comments with critical evaluation of the manuscript. There are concerns raised by the reviewers but I am happy to provide an opportunity for the reviewers to address concerns and submit a revision. Please carefully address the reviewers' comments.

·

Basic reporting

The paper successfully addresses a current and important issue of power allocation optimization in URLLC-D2D communication and presents an innovative approach within the framework of the Rician fading channel model. The robustness of the methodology and the consistency of the presented simulation results increase the scientific contribution of the work.

Experimental design

The study's methodological depth and the consistency of the simulation results support the proposed solution's effectiveness. Moreover, addressing the requirements of low latency and high reliability demonstrates that the paper has practical value for 5G networks.

Validity of the findings

The study's theoretical infrastructure and consistency of simulation results make it an important contribution to 5G networks. In particular, targeting low latency and high-reliability requirements increases the article's academic value.

Reviewer 3 ·

Basic reporting

The objective of the paper is not clear and the proposed method suggested in the article is used so many times.
The simulation results are noot convincing as in my previous comments I told to increas the simulation graph to analyse the effectiveness of the work. The added results are not convincing.
The contributions of the article is again questionable as therer are nymerious work already published.

The abtstract is not highlighting the complexity issue, and how the throughput of the frame work can affect the other performnace of the framework.

In previous paper, I have asd "parameters selection", I am nit convinced withe the answers statsing "based on other papers literature it was selected"; What is the standard parameters and why you have selected?.

The system model is not clear and not related with the parameters of optimization. The framework’s performance under large-scale D2D networks with numerous devices and varying mobility patterns is not explored in detail.

While the study focuses on Rician fading channels, which are common in line-of-sight environments, it may not adequately address other fading conditions (e.g., Rayleigh or Nakagami fading) encountered in non-line-of-sight scenarios.
The emphasis on optimizing power allocation may overlook other critical aspects of URLLC-D2D communication, such as interference management, resource block allocation, or user mobility.
While aiming to meet URLLC reliability and latency targets, the optimized power allocation might lead to increased power consumption, which is detrimental in energy-constrained devices.

The study may not account for 6G developments or dynamic spectrum sharing, which could alter the assumptions and relevance of the proposed model.

Experimental design

The objective of the paper is not clear and the proposed method suggested in the article is used so many times.
The simulation results are noot convincing as in my previous comments I told to increas the simulation graph to analyse the effectiveness of the work. The added results are not convincing.
The contributions of the article is again questionable as therer are nymerious work already published.

The abtstract is not highlighting the complexity issue, and how the throughput of the frame work can affect the other performnace of the framework.

In previous paper, I have asd "parameters selection", I am nit convinced withe the answers statsing "based on other papers literature it was selected"; What is the standard parameters and why you have selected?.

The system model is not clear and not related with the parameters of optimization. The framework’s performance under large-scale D2D networks with numerous devices and varying mobility patterns is not explored in detail.

While the study focuses on Rician fading channels, which are common in line-of-sight environments, it may not adequately address other fading conditions (e.g., Rayleigh or Nakagami fading) encountered in non-line-of-sight scenarios.
The emphasis on optimizing power allocation may overlook other critical aspects of URLLC-D2D communication, such as interference management, resource block allocation, or user mobility.
While aiming to meet URLLC reliability and latency targets, the optimized power allocation might lead to increased power consumption, which is detrimental in energy-constrained devices.

The study may not account for 6G developments or dynamic spectrum sharing, which could alter the assumptions and relevance of the proposed model.

Validity of the findings

The objective of the paper is not clear and the proposed method suggested in the article is used so many times.
The simulation results are noot convincing as in my previous comments I told to increas the simulation graph to analyse the effectiveness of the work. The added results are not convincing.
The contributions of the article is again questionable as therer are nymerious work already published.

The abtstract is not highlighting the complexity issue, and how the throughput of the frame work can affect the other performnace of the framework.

In previous paper, I have asd "parameters selection", I am nit convinced withe the answers statsing "based on other papers literature it was selected"; What is the standard parameters and why you have selected?.

The system model is not clear and not related with the parameters of optimization. The framework’s performance under large-scale D2D networks with numerous devices and varying mobility patterns is not explored in detail.

While the study focuses on Rician fading channels, which are common in line-of-sight environments, it may not adequately address other fading conditions (e.g., Rayleigh or Nakagami fading) encountered in non-line-of-sight scenarios.
The emphasis on optimizing power allocation may overlook other critical aspects of URLLC-D2D communication, such as interference management, resource block allocation, or user mobility.
While aiming to meet URLLC reliability and latency targets, the optimized power allocation might lead to increased power consumption, which is detrimental in energy-constrained devices.

The study may not account for 6G developments or dynamic spectrum sharing, which could alter the assumptions and relevance of the proposed model.

Additional comments

The objective of the paper is not clear and the proposed method suggested in the article is used so many times.
The simulation results are noot convincing as in my previous comments I told to increas the simulation graph to analyse the effectiveness of the work. The added results are not convincing.
The contributions of the article is again questionable as therer are nymerious work already published.

The abtstract is not highlighting the complexity issue, and how the throughput of the frame work can affect the other performnace of the framework.

In previous paper, I have asd "parameters selection", I am nit convinced withe the answers statsing "based on other papers literature it was selected"; What is the standard parameters and why you have selected?.

The system model is not clear and not related with the parameters of optimization. The framework’s performance under large-scale D2D networks with numerous devices and varying mobility patterns is not explored in detail.

While the study focuses on Rician fading channels, which are common in line-of-sight environments, it may not adequately address other fading conditions (e.g., Rayleigh or Nakagami fading) encountered in non-line-of-sight scenarios.
The emphasis on optimizing power allocation may overlook other critical aspects of URLLC-D2D communication, such as interference management, resource block allocation, or user mobility.
While aiming to meet URLLC reliability and latency targets, the optimized power allocation might lead to increased power consumption, which is detrimental in energy-constrained devices.

The study may not account for 6G developments or dynamic spectrum sharing, which could alter the assumptions and relevance of the proposed model.

·

Basic reporting

The last one or two paragraphs of the introduction should explain the contributions in detail followed by the organization of the remaining manuscript.
for reference see the end of the introduction in the following manuscripts.
doi: 10.1109/ACCESS.2021.3090965
doi: 10.23919/ChiCC.2017.8028558.

Experimental design

Unlike traditional cellular/Wi-Fi communication system, the d2d communication is driven by devices without the support of base station. in this content, the experimental design and the proper placement of proposed algorithm can be visually expressed to improve the clarity of solution, for reference you can see the figure 3 in the following manuscript.

doi: 10.1109/ACCESS.2021.3090965

Validity of the findings

The validity of the findings looks fine.
The Figures 2-6 in the manuscript have different size, they all should be adjusted to the same size to make the manuscript visually more appealing.

---

## Round 0.3 · accepted · Accept

Dear Authors,

I have assumed the role of academic editor. In the previous revision, one reviewer accepted your article. It was observed that one of the remaining two reviewers did not respond to the invitation within the expected time. In the current situation, the other reviewer has accepted your article. I have also conducted my own assessment of the revision, and I find it to be satisfactory. The paper now appears to be in a state suitable for publication.

Best wishes,

·

Basic reporting

ok

Experimental design

ok

Validity of the findings

ok

Additional comments

All the comments are addressed, and the manuscript should be accepted for publication.